# Long Non-coding RNAs Mechanisms of Action in HIV-1 Modulation and the Identification of Novel Therapeutic Targets

**DOI:** 10.3390/ncrna6010012

**Published:** 2020-03-13

**Authors:** Roslyn M. Ray, Kevin V. Morris

**Affiliations:** The Center for Gene Therapy, The Beckman Research Institute, City of Hope Medical Center, Duarte, CA 91010, USA; rray@coh.org

**Keywords:** HIV, lncRNA, transcription, antisense, epigenetic

## Abstract

This review aims to highlight the role of long non-coding RNAs in mediating human immunodeficiency virus (HIV-1) viral replication, latency, disease susceptibility and progression. In particular, we focus on identifying possible lncRNA targets and their purported mechanisms of action for future drug design or gene therapeutics.

## 1. Introduction

The interplay between host gene expression and exposure to pathogens is a vital process that occurs in vivo. Cellular fate is determined through the differential expression of key pathways that are regulated and fine-tuned, in part, by long non-coding RNAs (lncRNAs). Manipulation of these lncRNAs by pathogens, genomic rearrangements or single nucleotide polymorphisms (SNPs) have been shown to exert great effects on disease outcomes [1,2,3]. Long non-coding RNAs are an important class of RNAs involved in orchestrating and manipulating transcriptional and post-transcriptional effects in vivo. Further, lncRNAs have been found to undergo purifying selection, suggesting their functional importance in biological processes [4]. LncRNAs are RNAs that are over 200 nucleotides in length and usually have no functional coding potential [5]. LncRNAs work in *cis* or *trans*, or both, to exert their gene modulatory effects. These modulatory effects can occur in a myriad of ways, including: recruitment of chromatin remodeling factors to either repress or de-repress gene expression [6]; working as enhancers to facilitate promoter–enhancer interactions [7,8]; acting as an inhibitor to RNA polymerase II (RNAPII) binding through transcriptional interference [9]; acting as a scaffold for regulatory proteins at target sites in the DNA [10]; as competitive endogenous RNAs (ceRNAs), acting as microRNA sponges [11,12] or as decoys [13,14]; through facilitating mRNA splicing [15]; as well as manipulating target protein function by directing changes in phosphorylation states [16]; all of which result in a change in gene expression. With such distinct molecular mechanisms attributed to a small percentage of characterized lncRNAs, we are now only beginning to uncover the scope of how these regulatory molecules function in vivo.

Human immunodeficiency virus (HIV-1) is a virus that results in the development of acquired immune-deficiency syndrome (AIDS) in individuals if left untreated. Upon entry into CD4+ T cells, HIV-1 is able to integrate its genome into the host chromosome and replicate with a high efficiency [17]. A small percentage of CD4+ T cells undergo a process known as latency, whereby the integrated provirus is silenced, maintaining a persistent population or viral reservoir that is able to harbor HIV-1 indefinitely [18,19]. This subpopulation of CD4+ T cells remain a challenge to current treatment strategies that are only able to effectively target actively replicating cells [20]. Over the past decade, the role of lncRNAs in modulating HIV-1 has broadened our understanding of how host factors can maintain viral latency and facilitate HIV-1 replication in target host cells [21,22,23,24,25].

The already complicated interactions between host and viral proteins are now made more complex with discoveries of both host and viral lncRNAs that modulate disease outcomes. These observations may also provide insight into differences in susceptibility and disease progression within or between population groups or individuals, which may aid in identifying new treatment strategies for HIV-1. Of strategic importance are the mechanisms of action of these particular lncRNAs, which will provide researchers with novel interactions that can be therapeutically targeted. lncRNAs are attractive targets considering their expression is usually cell-type and disease-type specific [18,26,27,28]. Small molecule or oligonucleotide approaches that could silence or activate lncRNA expression are attractive treatment approaches that offer a high degree of target specificity.

In this review, we highlight new research into lncRNA mechanisms of action, new strategies to investigate lncRNAs in disease and possible strategies for the therapeutic targeting of HIV-1.

## 2. HIV-1 Transcribed Long Non-Coding RNA

LncRNA identification in HIV-1 regulation became prominent upon the discovery of an antisense transcript identified by several groups [29,30,31] originating from the *Nef* region in the HIV-1 genome (Figure 1). This lncRNA was, in fact, one of the first antisense lncRNAs discovered [32], the other being an antisense for MYC [33]. However, at the time of discovery the focus was on what protein these antisense RNAs were generating. Research into the function of this lncRNA suggests that it has an important role in the transcriptional control of HIV-1. Indeed, work from our group found that the HIV-1 antisense transcript originating from the *Nef* region of HIV-1, recruited a repressive complex consisting of EZH2 (enhancer of Zeste 2 polycomb repressive Complex 2), DNMT3a (DNA methyltransferase 3 alpha) and HDAC-1 (histone deacetylase 1) to mediate epigenetic silencing of HIV-1 at the 5′ long terminal repeat (5′LTR) [10]. This antisense transcript was highly enriched at the 5′LTR-gag region of HIV-1, and its subsequent knockdown resulted in significant de-repression of HIV-1. Data from our group [10] as well as others [34] suggest that this antisense transcript is involved in modulating HIV-1 latency through epigenetic silencing [10]. To further support this observation, Zapata et al. (2017) observed that cells expressing high levels of this transcript demonstrated a rapid return to latency after stimulation with latency reversal agents (LRAs) [35]. Importantly, this lncRNA was detected in HIV-1-positive patients using a novel biotinylated primer approach [36]. The authors detected this antisense transcript in activated CD4+ T cells and found that levels of the antisense transcript were lower in patients on antiretroviral (ART) therapy compared to untreated individuals [36]. Interestingly, this antisense transcript is predominantly found in active CD4+ T cells, while almost undetectable in resting CD4+ T cells [35,36]. It could be that this lncRNA is important in establishing latency.

To complicate the definitions of a lncRNA, this transcript has been postulated to be protein-coding, producing an antisense protein (ASP) (Figure 1) [37]. Recent evidence has shown that ASP is recognized by T cells, is evolutionarily conserved in the M group of HIV-1 [37,38], and has been found to be a transmembrane protein on the surface of the host cell part of the viral envelope, upon viral budding [39]. Taken together, the antisense transcript expressed in HIV-1-infected individuals, appears to work as a scaffold, directing epigenetic factors towards the 5′LTR of the HIV-1 promoter, contributing towards the establishment of latency. Furthermore, its potential protein (ASP) could form an integral part of the viral envelope structure. As such, the HIV-1 antisense lncRNA may be a useful target in which to prevent a return to latency after stimulation with latency reversal drugs. This could lead to more effective strategies to eliminate the viral reservoir.

## 3. Host-Transcribed Non-Coding RNAs Regulating HIV-1 Entry, Replication and Latency

The interaction of viruses and host factors has been well documented in the literature [10,40,41,42,43,44]. Recently, we have started to expand upon our understanding of host–virus interactions to include non-coding RNAs [21,45,46,47]. In particular, how viruses are able to dysregulate immune function has been a focal point. Several new studies investigating the roles of host lncRNA-HIV-1 interactions have revealed how HIV-1 is able to co-opt or suppress endogenous lncRNA mechanisms to regulate viral replication and infection. Further, recent studies have highlighted how lncRNAs are regulated in a time- [48] and cell-specific manner [49,50]. In HIV-1, some lncRNAs have been shown to have differential effects at different phases of the virus life cycle [51,52].

### 3.1. NEAT1

One such lncRNA is the nuclear-enriched abundant target 1 (NEAT1). NEAT1 has been found to be enriched and necessary for paraspeckle formation in the nucleus [53]. These paraspeckle bodies are integral to the internal organization of the nucleus and are responsible for the storage and transport of nuclear RNA, thereby regulating the expression of certain genes in vivo [52,53]. NEAT1 has been found to modulate HIV-1 expression in a post-transcriptional manner, by storing excess unspliced instability (INS)-containing HIV-1 RNA transcripts in paraspeckle bodies in the nucleus [52]. Thus, an increase in NEAT1 leads to an increase in INS-containing unspliced HIV-1 RNA transcripts stored in the paraspeckles, serving as a counter-balance HIV-1 transcription in the cell [52]. Zhang et al. (2013) postulated that these unspliced instability (INS)-containing transcripts can be transported in a HIV-1 Rev-dependent transport into the cytoplasm when needed, or can be modified by ADAR (Adenosine Deaminase RNA Specific) A-I RNA editing [52]. Budhiraja et al. (2015) determined that these paraspeckles containing the protein RBM14 are responsible for shuttling these NEAT1-associated unspliced RNA transcripts into the nucleolus that either transports the RNA into the cytoplasm or processes it for degradation [51]. These data further support the notion that NEAT1 facilitates the transport of HIV-1 transcripts into the paraspeckles for correct processing and storage. Interestingly, when HIV-1-infected donor T cells were stimulated with phytohemagglutinin [PHA), NEAT1 lncRNA expression was reduced, and the T cells were subsequently more susceptible to apoptosis [24]. The authors suggest that a reduction in NEAT1 may increase in HIV-1 replication, following the activation of resting CD4+ T cells [24]. Thus, NEAT1 RNA levels affect HIV-1 replication (Figure 2). The first way is through sequestration of unspliced RNA into paraspeckles which may aid in the regulation of viral transcription during active viral replication. The second is way through an unknown mechanism whereby NEAT1 is downregulated during viral reactivation from a resting state in CD4+ T cells, to promote HIV-1 transcription, and potentially HIV-1 dissemination [24,52]. This dynamic role of NEAT1 on HIV-1 replication and reactivation shows the importance of timing and context of lncRNA involvement in HIV-1 regulation.

### 3.2. NRON

The lncRNA, noncoding repressor of Nuclear Factor of Activated T cells (NFAT, or NRON), has been shown to have two distinct mechanisms of action in modulating HIV-1 transcription. Imam et al. (2015) found that NRON regulates HIV-1 transcription in a Nuclear Factor of Activated T cells (NFAT)-dependent manner [54]. NRON has been shown to be regulated by HIV-1 accessory proteins Nef and Vpu, suggesting that this lncRNA may be modulated differentially depending on the stage of the viral lifecycle. Downregulation of NRON, by Nef, increases NFAT protein levels, which results in an increase in HIV-1 transcription. Moreover, Vpu leads to a decrease in NRON expression and, thus, a decrease in HIV-1 transcription [46]. NRON works by regulating the nuclear trafficking of NFAT in the cells [54]; however, how Nef and Vpu regulate NRON expression is yet to be validated. Subsequently, Li et al. (2016) determined that NRON is able to contribute towards HIV-1 latency by inducing HIV-1 Tat degradation, in an NFAT-independent manner [55]. The authors observed that, when NRON was knocked down, HIV-1 Tat protein levels significantly increased, while the mRNA levels remained constant, suggesting a post-transcriptional regulatory mechanism. Indeed, the authors found that NRON specifically binds to the Tat protein and recruits ubiquitin/proteasome processing proteins (CUL4B and PSMD11), forming a complex targeting Tat for degradation. The authors suggest that NRON is involved in maintaining latency, through the regulation of Tat in resting CD4+ T cells [55]. Thus, it appears that NRON, using two distinct mechanisms of action, is able to modulate HIV-1 post-transcriptionally through Nef and Vpu, and to maintain HIV-1 in a latent state through Tat degradation (Figure 2).

### 3.3. GAS5

The lncRNA growth arrest-specific transcript 5 (GAS5) inhibits HIV-1 replication by acting as a competing endogenous RNA (ceRNA), suppressing the effects of miR-873 [12]. GAS5 has been shown to be downregulated upon HIV-1 infection in T cells [54] and, using a combination of knockdown and overexpression strategies of GAS5, Chen et al. (2018) determined that high levels of GAS5 decreased HIV-1 transcription, while a knockdown of GAS5 resulted in an increase in HIV-1 transcription. Using a microRNA-RNA predicative interaction software, the authors found and confirmed experimentally that GAS5 interacts with miR-873, and that miR-873 increases HIV-1 transcripts in the cell [12]. However, whether this occurs through increasing mRNA stability of the HIV-1 transcripts, or if this miRNA indirectly regulates proteins involved in HIV-1 transcription has yet to be elucidated.

### 3.4. HEAL

Chao et al. (2019) identified lncRNAs in HIV-1-infected monocyte-derived macrophages (MDMs) three days post infection with HIV-1_BaL_ [56]. In particular, the authors focused on lncRNAs that were upregulated upon viral infection, and found that LINC02574-201 or HIV-1 enhanced lncRNA (HEAL) was significantly elevated upon infection in both MDMs and in primary CD4+ T cells. Interestingly, HEAL expression was absent in bystander and uninfected MDMs and CD4+ T cells, suggesting that viral infection upregulates HEAL [56]. Thereafter, the authors found that HEAL regulated HIV-1 transcription by acting as a scaffold to recruit fused in sarcoma (FUS) RNA-binding protein. This RNA–protein complex was found to be enriched at both the CDK2 promoter and at the integrated HIV-1 LTR, where this complex resulted in the recruitment of p300, through FUS, increasing transcription of both HIV-1 and CDK2 [56]. CDK2 itself, has been found to increase HIV-1 transcription through phosphorylation of multiple proteins essential for HIV-1 transcription: CDK9 [57], part of the positive transcription elongation factor (P-TEFb) complex, the viral accessory protein, Tat [58], and RNA polymerase II [59]. HEAL mediated these transcriptional effects by directly binding to the HIV-1 LTR (Figure 2) [56]. Interestingly, when this lncRNA was knocked down through CRISPR/Cas9, HIV-1 reactivation was prevented, despite withdrawal of azidothymidine (AZT) treatment in T cells and in microglial cells [56]. HEAL may be an attractive therapeutic target to prevent HIV-1 latency, especially considering that it is only upregulated in infected CD4+ and macrophages. Indeed, this lncRNA is also an attractive target for HIV-1 transcription. Considering that this lncRNA appears to function as a scaffold, it may prove interesting to determine those regions necessary for either DNA-lncRNA binding or protein recruitment and possibly, by using small molecule screens or small oligonucleotides complementary to the lncRNA region, abrogate the lncRNA function in vivo.

### 3.5. MALAT1

Qu et al. (2019) identified that the known cancer-associated lncRNA, metastasis-associated lung adenocarcinoma transcript 1 (MALAT1), increased HIV-1 transcription, through altering the epigenetic status of the HIV-1 promoter. Interestingly, the authors found that MALAT1 acted as a sponge, limiting the amount of EZH2 from the polycomb repressive complex 2 (PRC2) [60]. This resulted in a loss of PRC2-mediated methylation at the LTR promoter and, thus, a reduction in the epigenetic silencing of HIV-1. Further, the authors observed that MALAT1 expression increased when ACH2 cells were treated with latency reversing agents (LRAs), suggesting that MALAT1 may contribute to viral reactivation [60]. Interestingly, the authors found that MALAT1 was highly expressed in activated T cells compared to un-activated T cells, suggesting that MALAT1 may play a role in initial viral replication, as well as in disease progression, through modulating the epigenetic status of HIV-1 [60].

### 3.6. LINC01426

Huan et al. (2018) investigated the role of the lncRNA *uc002yug.2* or *LINC01426* in HIV-1 replication and reactivation. This research showcases how a single lncRNA can interact with both host and viral factors to mediate viral transcription. LINC01426 enhanced HIV-1 replication in a post-transcriptional manner [61]. An increase in LINC00146 resulted in a switch in the predominant isoforms of the transcription factor, runt-related transcription factor 1 (RUNX1), decreasing RUNX1b/c, and increasing the RUNX1a isoform [62]. These transcription factors are able to bind to the 5′ LTR of HIV-1 at a RUNX1 DNA-binding domain in the U3 region [62]. This switch to a predominant RUNX1a phenotype increases HIV-1 replication. In addition, this leads to an increase in Tat expression and a possible LINC01426-Tat interaction that further increases HIV-1 replication (Figure 2). The authors found that the lncRNA is also increased by HIV-1, and decreased in latent cell lines as well as in HAART (highly active antiretroviral therapy)-treated patients’ CD4+ T cells. Additionally, LINC01426 may have a role in viral reactivation through its interaction with the viral protein Tat. Interestingly, the authors noted that the effects were not always observed or as pronounced in certain cell lines and donors [62]. Further research is needed to understand the heterogeneous effects of this lncRNA on HIV-1 modulation. It may be that a putative second isoform of LINC01426 (as observed in the UCSC genome browser; Human Dec. 2013 (GRCh38/hg38) Assembly) may play a role in modulating the predominant isoform in cell lines or, in the case of primary cells, an SNP in the lncRNA itself that may alter its function in vivo.

### 3.7. LINC00173

Postler et al. (2017) screened RNA-sequencing (RNA-seq) data to identify lncRNAs regulated at early HIV-1 infection time points (12 and 24 h) in a T cell line. Of interest was an lncRNA with two transcript variants, *LINC00173*. The authors found that this lncRNA was decreased in a dose- and time-dependent manner upon HIV-1 infection; however, the expression levels of the lncRNA had no observable effect on HIV-1 replication [25]. Rather, LINC00173 variants appeared to regulate a subset of cytokines, including interferon gamma (IFN-γ), C-C motif chemokine ligand 3 (CCL3) and C-X-C motif chemokine ligand 8 (CXCL8), with knockdown of the lncRNAs resulting in an increase in these cytokines. The authors postulated that HIV-1 co-opts expression of this lncRNA to dysregulate the immune response in a beneficial manner to promote HIV-1 replication and infection [25].

The lncRNAs reported herein underscore the diversity of lncRNA mechanisms and illustrate how these mechanisms impede or augment HIV-1 replication or viral latency. If we are to identify useful targets for HIV-1 therapeutics, it may be essential to perform comparative analyses of these lncRNAs on HIV-1 replication and latency. For example, several of the lncRNAs presented in this section may be viable targets for HIV-1 replication control. HEAL [56], MALAT-1 [60] and LINC01426 [62] could be targeted directly through RNA interference to decrease their expression and limit HIV-1 replication. However, it is also imperative to determine the broad regulatory effects of these lncRNAs upon knockdown conditions. Knockdown of these lncRNAs in other cell types may have unintended consequences. Thus, it is necessary to screen these lncRNAs in terms of their distinct effects on HIV-1 transcription and or latency, as well as the effects of these lncRNAs in normal cell function. Alternatively, one could test small molecules that may specifically inhibit lncRNA binding to its target. However, toxicity and efficacy must also be determined.

## 4. Host-Transcribed Non-Coding RNAs Regulating HIV-1 in a Cell-Specific Manner

LncRNAs have shown to have distinct cellular expression patterns, suggesting that lncRNA-HIV-1 interactions need to be viewed in the context of the cell type. However, no studies have directly compared different target cell (resting CD4+ T cells, activated CD4+ T cells, or macrophages) lncRNA expression profiles upon HIV-1 infection at different time points and determined whether there are unique lncRNA signatures. However, several new studies have focused on lncRNA expression profiles in individual cell types.

### 4.1. LincRNA-p21

Barichievy et al. (2018) sought to understand the mechanism by which HIV-1-infected macrophages evade apoptosis, despite DNA damage mediated through HIV-1 integration [63]. The authors found that LincRNA-p21, is dysregulated upon initial viral infection in an envelope glycoprotein-120 (gp-120) and extracellular signal-regulated kinase 2 (ERK2)-dependent manner. This results in the complexation of the host protein human antigen R (HuR) and the lncRNA in the nucleus, which results in the degradation of the lncRNA. By diverting the lincRNA-p21 to associate with HuR, the host protein heterogeneous nuclear ribonucleoprotein K (hnRNP-K), is sequestered into the cytoplasm. This sequestration of hnRNP-K prevents the formation of the nuclear pro-apoptotic complex (LincRNA-p21- hnRNP-K). This results in an increase in expression of pro-survival genes and, thus, HIV-1 is able to evade apoptosis in macrophages. Included in this repertoire of pro-survival genes is mitogen-activated protein kinase kinase 1 (MAP2K1). Increased MAP2K1 ensures that the pro-survival cascade is maintained in macrophages, through the sustained phosphorylation of ERK2. In addition, ERK2 phosphorylation leads to the degradation of hn-RNPK in the cytoplasm, and incomplete activation of the transcription factor p53. Partial activation of p53 leads to a dysregulation of p53-dependent genes, including Linc-p21. As a consequence, Linc-p21 expression levels are reduced (see Figure 3) [63]. This is incredibly complex mechanism demonstrates how HIV-1 is able to efficiently hijack host mechanisms to control the cellular response towards its infection. Interestingly, this mechanism of evasion is absent in CD4+ T cells.

Furthermore, the authors used novel small molecules to prevent hnRNP-K sequestration into the cytoplasm of infected macrophages, and found that the addition of the small molecule reversed the HIV-1-induced anti-apoptotic effects, resulting in cell death [63]. Thus, determining the mechanism of action of lncRNAs is crucial to identifying key components in the pathway that can be manipulated therapeutically. Further, this paper exploits already identified small molecules to mediate their desired therapeutic effect. As such, it may be that new molecules need not be identified if we know which mechanism we need to target.

### 4.2. FAS Antisense 1 (FAS-AS1)

Boliar et al. (2019) identified lncRNAs that were differentially expressed in monocyte-derived macrophages (MDMs) infected with HIV-1 [64], and focused on how these lncRNAs determine cellular fate upon HIV-1 infection. The authors observed that the antisense lncRNA, FAS antisense 1 (FAS-AS1) was significantly increased in infected MDMs compared to bystander or uninfected MDMs. The authors found that repression of FAS-AS1 lead to an increase in caspase-3/7 activity and, thus, an increase in apoptosis mediated through FAS (Fas cell surface death receptor). Thus, FAS-AS1 appears to prevent apoptosis in macrophages infected with HIV-1 [64]. SAF decreases apoptosis in a post-transcriptional manner, through induction of alternative splicing of FAS mRNA [65]. Interestingly, FAS-AS1 does not appear to have these effects in T cells [52] and, thus, these effects may be constrained within macrophages [64]. Further, these data suggest that the differences in lncRNA expression may contribute towards the different cellular fates observed in these cell types upon HIV-1 infection.

Both lncRNAs described in this section indicate how different lncRNA signatures in different cell types contribute towards the phenotypes observed upon HIV-1 infection in macrophages and T cells [64,65]. These data support the idea that regulation of lncRNAs by HIV-1 should be viewed within different cellular contexts to gain a more meaningful insight into differences between target cell types. For example, Hudson et al. (2019) investigated lncRNA expression profiles in both human and mouse T cell subsets infected with lymphocytic choriomeningitis virus [66]. The authors found that there were distinct patterns of polyadenylated lncRNAs that could discriminate between CD8+ naïve, effector and memory T cell subsets. Over a quarter of the RNAs transcribed were non-coding, in both mice and human CD8+ T cell subsets, showing the significant contribution of non-coding RNAs in the expression repertoire of T cells [66]. Data generated by these groups [63,64,66] indicate that it would be beneficial to stratify lncRNA screens in T cells and macrophages. Further, it would be useful to delineate between the different phases of the viral life cycle—infection, replication and latency. These datasets may provide us with unique cellular signatures that could offer new treatment targets. This is particularly intriguing when considering the latently infected resting CD4+ T cells and reactivated virally replicating CD4+ T cells. If unique signatures could be identified that result in the reactivation of the virus, this offers a bigger molecular toolkit with which to manipulate reactivation for HIV-1 treatment or to potentially eliminate the viral reservoir.

## 5. LncRNAs and HIV-1 Susceptibility

Very few papers have identified the roles of lncRNAs in HIV-1 susceptibility. In the first study to identify an lncRNA associated with HIV-1 susceptibility, Kulkarni et al. (2019) identified single nucleotide polymorphisms (SNPs) and their disease outcomes associated with HIV-1 using previous genome-wide association studies (GWAS) [67]. Here, the authors found that an SNP (rs1015164A/G), altered the binding capacity of the lncRNA C-C chemokine receptor 5 antisense (CCR5AS), with the host RNA-binding protein, Raly. The G mutation resulted in a higher CD4+ T cell count and lower viral load in HIV-1-positive patients over time. C-C chemokine receptor 5 (CCR5) is critical for R-tropic HIV-1 viral entry [68], as such, changes in the expression levels of this co-receptor will greatly affect susceptibility and disease progression. The authors found that, in the absence of the lncRNA, Raly is able to bind to the 3′ untranslated region (UTR) of CCR5 and process the mRNA for degradation. Further, a second SNP rs2027820A/G altered the activating transcription factor 1 (ATF1)-binding site in the antisense lncRNA *CCR5AS* gene, which leads to either the high binding capacity (G) of the transcription factor ATF1, or a reduced binding capacity (A). These changes in ATF1 binding, lead to either higher (G) or lower (A) levels of CCR5AS lncRNA (see Figure 4). Further the authors found that these SNPs exhibit linkage disequilibrium, meaning that these SNPs often occur together.

Taken together, the authors observed that SNPs in the *CCR5AS* gene resulted in a change in HIV-1 infection levels due to differences in CCR5 protein levels in target cells. In particular, the authors concluded that rs1015164A is associated with a lack of HIV-1 control, due to the higher levels of CCR5 on target T cells [67]. Thus, SNP variations in lncRNAs can greatly alter the susceptibility of an individual to a disease.

Like Kulkarni et al. (2019), a growing number of research has focused on SNPs within lncRNAs, and their subsequent effect on lncRNA function and disease outcomes [69,70,71,72]. Hua et al. (2018) determined that an SNP in the lncRNA *PCAT19* (Prostate Cancer Associated Transcript 19) promoter, could alter expression of the lncRNA isoforms to promote prostate cancer [73]. SNPs within the lncRNA *H19* have been found to have an increased risk of hepatocellular [61] and gastric cancer [72]. The lncRNA *HOTAIR* (HOX Transcript Antisense Intergenic RNA) has multiple SNPs associated with an increased risk of several cancer types [69,70,74,75]. These SNPs may affect the expression of the lncRNA or affect its secondary structure [67,73] which alters its canonical function in the cell [76]. Thus, identifying disease-associated SNPs within lncRNAs associated with HIV-1 replication, could provide insight into differences in susceptibility and disease progression observed within certain individuals. What would prove highly insightful is a comparison of highly exposed but seronegative (HESN) individuals compared to HIV-1-infected and HIV-1-negative individuals to determine SNP differences that may contribute towards the HESN profile. The addition of several recently updated lncRNA-SNP databases, which identify possible lncRNA and their associated SNPs in complex disease traits [77,78], may help identify new key lncRNAs involved in HIV-1 modulation, and, alongside this, identify new markers of risk as well as potential therapeutic targets.

## 6. LncRNAs Involved in Disease Progression or Secondary Disease Outcomes

The role of lncRNAs in HIV-1 is not limited to direct effects on viral replication, but also to their contribution to secondary disease outcomes. Recent studies have highlighted the roles of lncRNAs in mediating different disease outcomes.

For example, Chinnappan et al. (2019) investigated the role of HIV-1 and cocaine use in the development of HIV-1 pulmonary arterial hypertension (PAH) [46]. Viral proteins, like Nef and Tat [79,80], as well as co-treatment with cocaine [81] have previously been observed to result in the increased proliferation of smooth muscle cells, which leads to the downstream development of PAH. Chinnappan et al. (2019) determined the lncRNA–microRNA interactions upon HIV-1 Tat and cocaine exposure in primary human pulmonary arterial smooth muscle cells (HPASMCs) [46]. The authors stratified their analysis to determine which lncRNA–miRNAs, lncRNA–mRNAs and lncRNA–mRNA–miRNAs were most likely associated with their disease outcome using multiple types of pathway analysis software [46]. By parsing out and stratifying for lncRNAs that had mRNA and miRNAs associations, the authors were able to identify several cis-acting lncRNAs and ceRNAs that may contribute towards smooth muscle cell development upon HIV-1 Tat and cocaine co-exposure. The authors identified two lncRNAs in modulating smooth muscle cell proliferation (ncRNA ENST00000495536 associated the HOXB13 gene expression and lncRNA ENST00000585387, a ceRNA for miRNAs, miR-491 and miR-185) [46] and found that, when these lncRNAs were silenced using siRNA, the effects of cocaine and Tat treatment on cell proliferation were lost, suggesting that these lncRNAs may play a role in the development of PAH in HIV-1-positive individuals [46].

Zhou et al. (2018) investigated the role of acute exposure of the viral protein Nef in regulating lncRNAs in mouse astrocytes, as a proxy to the development of HIV-1-associated neurological disorder (HAND) [82]. HAND is associated with persistent inflammation in the brain. In particular, they found that the lncRNA AK006025 increased a select repertoire of cytokines associated with inflammation, CXCL9, CXCL10 (IP-10), and CXCL11 in astrocytes upon stimulation with soluble Nef protein. Further, they suggested that these effects occurred through an lncRNA–protein interaction with the transcription factor CREB (cAMP response element-binding protein) binding protein (CBP/P300) and the nuclear factor kappa-light-chain enhancer of activated B cells (NFκB) p65 isoform. This association/and or interaction of AK006025 with CBP/P300 and NFκB p65, resulted in an enrichment of CBP/P300 and NFκB p65, as well as an increase in acetylation (H3K27ac), a marker for an active promoter, at the promoter sites of *CXCL9*, *CXCL10*, and *CXCL11* genes [82]. However, the authors did not elucidate the full mechanism of this lncRNA. It has yet to be determined whether there is a direct association with this lncRNA at these promoter sites, or if the lncRNA modulates the epigenetic landscape of the promoters, or whether it acts at a post-transcriptional level, working to recruit proteins to activate CBP/P300 and NFκB. However, due to the poor conservation of lncRNAs between species, it would be beneficial to determine the effect in human astrocytes. Nevertheless, the paper demonstrates the effects of HIV-1 accessory proteins, such as Nef [82] on the non-coding transcriptome, and how this could contribute towards secondary disease outcomes associated with HIV-1 disease progression.

These articles present thoughtful methodological approaches in screening for lncRNA function. In particular, the authors were able to screen lncRNAs for their relevancy in their bioinformatic approaches. This aided in the identification of lncRNAs with the most pronounced effects in their disease models [46,82]. Further, Chinnappan et al. (2018) identified lncRNAs with potential direct effects on their mRNA targets, by screening for lncRNA–mRNA and lncRNA–mRNA–miRNA associations [46]. This may be a useful approach in light of therapeutic targeting.

## 7. LncRNAs as Therapeutic Targets for HIV-1

LncRNAs are attractive targets for therapeutic interventions due to their tissue and cell-specific nature [83]. Several methods to investigate lncRNAs of biological importance have been utilized to great effect by the papers reported herein, and have highlighted how critical it is to have focused studies if we are to identify key lncRNAs as candidates for therapeutic targeting. One approach would be to re-evaluate HIV-1 GWAS datasets to identify lncRNA–SNP interactions that result in disease-associated traits. Additionally, one could perform RNA-seq analyses at the different phases of HIV-1 infection in target human T cells that would aid in identifying unique signatures. For example, Chen et al. (2019) determined the lncRNA–mRNA co-expression profiles in elite controllers versus normal-process HIV-1 patients [22], and found that these profiles were differentially enriched in immune response-regulating signaling and apoptosis pathways [22]. Alternatively, one could screen based on lncRNA mechanism. Tan et al. (2017) screened specifically for *cis*-acting lincRNAs (long-intergenic non-coding RNA) associated with complex disease traits in topologically associated domain (TAD) boundaries in human lymphoblastoid cells [84]. Mirza et al. (2014) identified lncRNA–SNP variants in type 1 diabetes and inflammatory bowel disease, and specifically sought to identify if SNPs resulted in the disruption of the lncRNA secondary structure that could be a potential feature for disease outcome [76]. Chinnappan et al. (2019) primarily focused on identifying ceRNAs through lncRNA–miRNA screenings [46]. Whatever the choice, these targeted approaches could be highly beneficial if we want to identify novel targets for HIV-1 treatment. The hope is to identify cell-specific lncRNAs involved in HIV-1 replication or latency that are targetable. This will help the developed therapeutic method to be both effective and have a low off-target profile in vivo.

Of critical importance in all these approaches is the accuracy of the lncRNA annotations currently available. Currently, NONCODE v5 [85], GENCODE v7 [86] and LNCipedia v5 [87] offer some of the most comprehensive lncRNA annotations [88]. Importantly, as Uszczynska-Ratajczak et al. (2018) described in their review, most databases are enriched for polyadenylated fractions, with lowly expressed lncRNAs underrepresented by most sequencing techniques [88]. However, they note that, with recent advances in RNA CaptureSeq and sequencing technology, more accurate, sensitive and reliable annotations are now achievable [88]. Thus, we may be able to identify important lncRNAs associated with HIV-1 susceptibility, replication and latency. By understanding their molecular mechanisms of action, we may then be able to develop specific and targeted therapeutic approaches in the treatment of HIV-1. As stated previously, one could also comparatively test the already identified lncRNAs reported herein. The identification of the lncRNA with the most pronounced effect on HIV-1 replication and or latency could then be exploited for therapeutic targeting, whereupon the next steps would involve determining the best mode of targeting (small molecule versus RNAi), safety evaluations (including off-target effects) and delivery.

The identification of the mechanism of action is critically important for the development of small molecule therapeutics that could disrupt these key interactions in vivo [63]. Recent advances into small molecules that interact specifically with RNA species suggest that relatively rapid identification and screening are now achievable [89,90]. The key to this approach is to not only determine the mechanism of action of the lncRNA, but to determine what regions are critical for its mechanism. This approach still requires advanced techniques to identify potential small molecules [89,90]. The benefit, however, is the development of an lncRNA-specific small targeting molecule.

Alternatively, the use of RNAi technology could be used to target lncRNAs for knockdown [91,92,93,94,95,96]. RNAi, including antisense oligonucleotides, gapmeRs and siRNAs, are effective tools for gene therapy due to their ease of screening and upscaling. Current RNAi-based therapeutics have been developed with some success, with several RNAi-based therapies approved by the Federal Drug Administration (FDA) [97]. However, as reviewed by Win and Rogge et al. (2019), toxicity profiles of these modalities need to be improved [97]. Of promise are the recent clinical trials using conjugated siRNAs to treat cholesterol, whereby a single dose of the targeted siRNA molecules had lasting effects on in vivo with minimal side effects reported [98,99].

While delivery into target cells or tissue types has not been extensively explored with regards to HIV-1, the use of nanoparticles [92,100,101], liposomes [101,102,103,104], cell-penetrating peptides [105,106], receptor-mediated targeting [107,108,109] and aptamers [110,111,112] have been assessed for use in cell based-therapeutic approaches. Further, results from our lab suggest that lncRNAs can be successfully targeted using either siRNA conjugates [113] or antisense oligonucleotides encapsulated in either nanoparticles or exosomes [114] to potentially mediate positive therapeutic outcomes. Further research is needed to identify the optimal delivery modalities that can target HIV-1 cells systemically and within target tissue sites.

## 8. Conclusions

The comprehensive research to date, highlighted in this review, demonstrates the plethora of lncRNAs utilized by HIV-1 at every stage of the viral infection cycle. The crosstalk that exists between the host cell and the HIV-1 virion results in a myriad of changes with the lncRNA transcriptomic landscape, which we are only beginning to unravel. Importantly as we continue to discover the mechanisms by which these lncRNAs exert their modulatory effects, we will discover novel targets for HIV-1 prevention and treatment.

## Figures and Tables

**Figure 1 ncrna-06-00012-f001:**
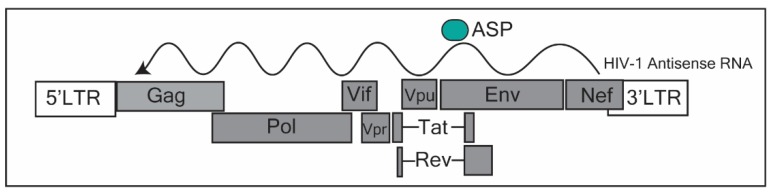
Annotation of the human immunodeficiency virus (HIV-1) genome. The HIV-1 antisense RNA transcriptional start site occurs in the *Nef* region. The putative protein, the antisense protein (ASP), is translated near the 5′ region of the antisense transcript, corresponding to the *Env* region in the HIV-1 genome. This figure was adapted from Saayman et al. (2014) [10] and Cassan et al. (2016) [37].

**Figure 2 ncrna-06-00012-f002:**
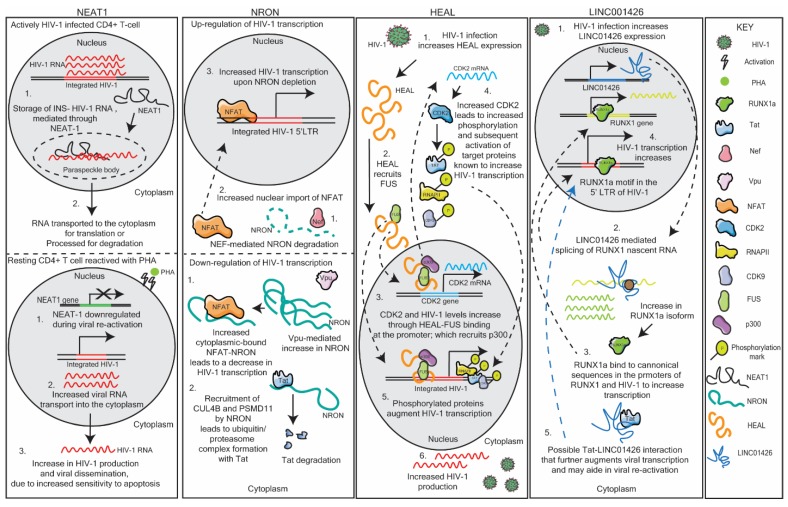
Schematic representation of the lncRNAs nuclear-enriched abundant target 1 (NEAT1), noncoding repressor of Nuclear Factor of Activated T cells (NRON), HIV-1 enhanced lncRNA (HEAL) and LINC01426 and their respective mechanisms of action in modulating HIV-1 replication. See text for details.

**Figure 3 ncrna-06-00012-f003:**
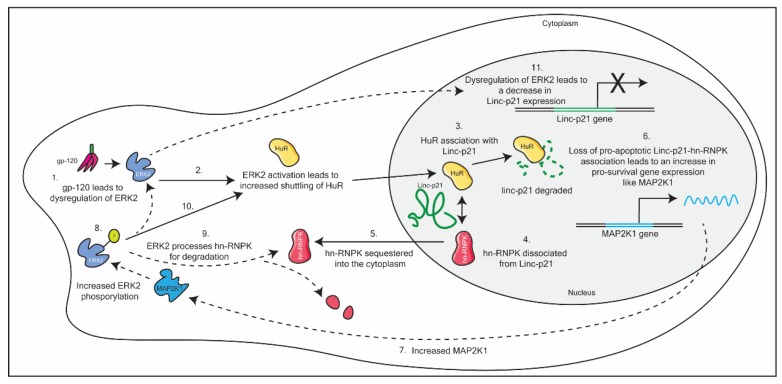
A schematic detailing HIV-1 dysregulation of Linc-p21 in macrophages. This dysregulation ultimately results in the evasion of apoptosis in macrophages, through a multi-step pathway. See text for details.

**Figure 4 ncrna-06-00012-f004:**
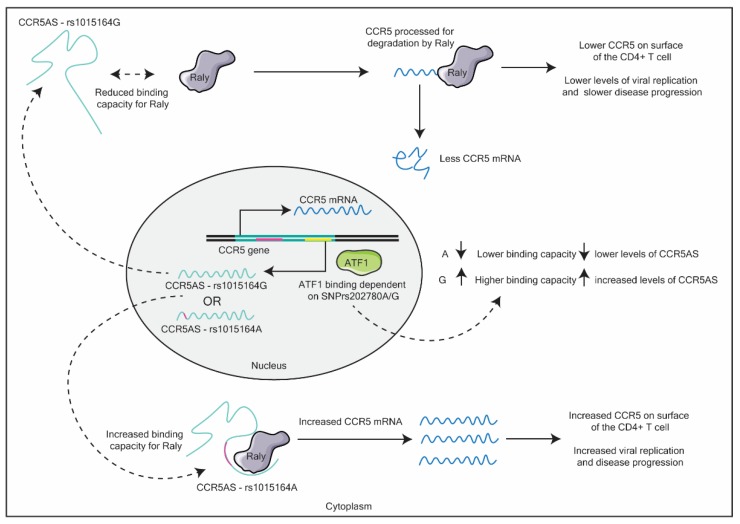
Schematic detailing the effects SNPs rs1015164(A/G) and rs202780(A/G) in the C-C chemokine receptor 5 antisense (*CCR5AS*) gene have on subsequent C-C chemokine receptor 5 (CCR5) mRNA levels. See text for details.

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
