# Peer review of "Long Non-coding RNAs Mechanisms of Action in HIV-1 Modulation and the Identification of Novel Therapeutic Targets"

_ncrna, 2020, doi:10.3390/ncrna6010012_

Round 1

Reviewer 1 Report

While the subject matter is important with regard to lncRNA control of HIV replication and their potential for therapeutic intervention, it is very poorly written. I have made extensive alterations to the text throughout, and would suggest the authors consider these before resubmitting.

Author Response

We thank the reviewers for their time and their valuable input in making this manuscript more thoughtful and scientifically rigorous. Thank you for accepting the manuscript for publication into ncRNA. We have made significant improvements to this manuscript and hope that these changes are acceptable.

Reviewer #1:

Comments and Suggestions for Authors

While the subject matter is important with regard to lncRNA control of HIV replication and their potential for therapeutic intervention, it is very poorly written. I have made extensive alterations to the text throughout, and would suggest the authors consider these before resubmitting.

Response

We have made significant improvements to the language and format of the manuscript. We have edited as per the reviewer’s suggestions, and hopefully this has improved the manuscript.

Reviewer 2 Report

Ray et al. review the involvement of lncRNAs in HIV-1 biology. The topic is interesting and timely and has not been the subject of regular literature reviews. The content is complete, but the organisation could be improved. A further weakness is that the authors often discuss the literature paper by paper, rather than providing a true synthesis.

I would support publication of this review upon modification by the authors.

Major points:

- The most obvious deficiency in this review is lack of figures. The authors must include figures that synthesise each major section.

- HIV-1 transcribed long non-coding RNAs: it is not clear whether the lncRNA discussed in the first paragraph is the same RNA as the one coding for ASP. Cassan et al. (ref 38) suggests there are different antisense RNAs, but the text hints that there is only one. A figure would really help here.

- Organisation of the host transcribed non-coding RNAs: I am not convinced that splitting host lncRNAs into two sections positive and negative regulators is helpful for comprehension, especially as many lncRNAs have pleiotropic effects. I would recommend one section 'host transcribed mRNAs' with subsections for each example i.e. NEAT1, NFAT etc. Alternatively, the authors could separately discuss role of lncRNAs in HIV-1 susceptibility to infection, replication, latency and disease progression.

- Precision of language: the authors are imprecise in their language in many places. The manuscript would be improved by carefully distinguishing effects of lncRNAs on transcription vs latency vs other stages of the replication cycle.

A good example is p3, l105, where the authors talk about how NEAT1 "aid(e) in the regulation of viral transcription". The author's had just discussed how NEAT1 regulates the transport of unspliced RNA into paraspeckles, which - strictly speaking - has nothing to do with 'transcription'. A better description is that NEAT1 regulates expression of unspliced HIV-1 mRNA post-transcriptionally. There are many places in the text where the authors need to carefully revise the text e.g. p2, l86; p3, l106-110 and elsewhere

Minor points:

- Abstract could be more focused, shorter and to the point.

- Abbreviations are often not defined upon first use, or the abbreviations are not used in later passages

- Consistency between the use of non-coding and long non-coding

- p1, l32: revise to "...integrate *its genome into the host chromosome* and..."

- p2, l52: Ref 30 is incorrect

- p2, l65: in in

- p2, l73, and elsewhere in the manuscript: remove "protein" from "protein scaffold"

- p3, l93: "as a counter-balance HIV-1 transcription in the cell" - this is unclear. Revise.

- p3, l94: Sentence beginning "Budhiraja et al. (2015)..." is unclear. Revise.

- p3: Why is TAT capitalised? Also NEF elsewhere

- p4, l163-178: this text seems out of place, as it does not deal with HIV-1 infection. It would work better as a discussion point later or remove.

-8, l372: revise to "...recent advances in RNA sequencing, such as CaptureSeq, ..."

Author Response

We thank the reviewers for their time and their valuable input in making this manuscript more thoughtful and scientifically rigorous. Thank you for accepting the manuscript for publication into ncRNA. We have made significant improvements to this manuscript and hope that these changes are acceptable.

Reviewer #2:

Comments and Suggestions for Authors

Ray et al. review the involvement of lncRNAs in HIV-1 biology. The topic is interesting and timely and has not been the subject of regular literature reviews. The content is complete, but the organisation could be improved. A further weakness is that the authors often discuss the literature paper by paper, rather than providing a true synthesis.

I would support publication of this review upon modification by the authors.

Major points:

- The most obvious deficiency in this review is lack of figures. The authors must include figures that synthesise each major section.

- HIV-1 transcribed long non-coding RNAs: it is not clear whether the lncRNA discussed in the first paragraph is the same RNA as the one coding for ASP. Cassan et al. (ref 38) suggests there are different antisense RNAs, but the text hints that there is only one. A figure would really help here.

- Organisation of the host transcribed non-coding RNAs: I am not convinced that splitting host lncRNAs into two sections positive and negative regulators is helpful for comprehension, especially as many lncRNAs have pleiotropic effects. I would recommend one section 'host transcribed mRNAs' with subsections for each example i.e. NEAT1, NFAT etc. Alternatively, the authors could separately discuss role of lncRNAs in HIV-1 susceptibility to infection, replication, latency and disease progression.

- Precision of language: the authors are imprecise in their language in many places. The manuscript would be improved by carefully distinguishing effects of lncRNAs on transcription vs latency vs other stages of the replication cycle.

A good example is p3, l105, where the authors talk about how NEAT1 "aid(e) in the regulation of viral transcription". The author's had just discussed how NEAT1 regulates the transport of unspliced RNA into paraspeckles, which - strictly speaking - has nothing to do with 'transcription'. A better description is that NEAT1 regulates expression of unspliced HIV-1 mRNA post-transcriptionally. There are many places in the text where the authors need to carefully revise the text e.g. p2, l86; p3, l106-110 and elsewhere

Minor points:

- Abstract could be more focused, shorter and to the point.

- Abbreviations are often not defined upon first use, or the abbreviations are not used in later passages

- Consistency between the use of non-coding and long non-coding

- p1, l32: revise to "...integrate *its genome into the host chromosome* and..."

- p2, l52: Ref 30 is incorrect

- p2, l65: in in

- p2, l73, and elsewhere in the manuscript: remove "protein" from "protein scaffold"

- p3, l93: "as a counter-balance HIV-1 transcription in the cell" - this is unclear. Revise.

- p3, l94: Sentence beginning "Budhiraja et al. (2015)..." is unclear. Revise.

- p3: Why is TAT capitalised? Also NEF elsewhere

- p4, l163-178: this text seems out of place, as it does not deal with HIV-1 infection. It would work better as a discussion point later or remove.

-8, l372: revise to "...recent advances in RNA sequencing, such as CaptureSeq, ..."

Response

Thank you for your valuable input and excellent recommendations considering the format of this review.

We have made significant improvements to the language and format of the manuscript. We have edited as per the reviewer’s suggestions, including the re-arrangement of the sections to include titles for each of the lncRNAs described.  We have also included figures detailing the mechanism of action for select lncRNAs. We hope that we have improved upon the accuracy and the synthesis of this review.

Round 2

Reviewer 1 Report

The revised manuscript is significantly improved over the previous version. Figures are easy to follow and the overall organization has improved. I recommend acceptance in the current form

Reviewer 2 Report

The author's have obviously worked hard on the manuscript: it is vastly improved and I would be a very useful contribution to the literature.

There are a few very minor grammatical errors (trailing commas, abbreviation use) that can be corrected upon final proof-reading.